# The experience of people living with heart failure in Ethiopia: A qualitative descriptive study

Henok Mulugeta[1,2]*, Peter M. Sinclair[2], Amanda Wilson[2]

**1** Department of Nursing, College of Health Sciences, Debre Markos University, Debre Markos, Amhara Region, Ethiopia, **2** School of Nursing and Midwifery, Faculty of Health, University of Technology Sydney, Sydney, New South Wales, Australia

* mulugetahenok68@gmail.com

## Abstract

### Background

Heart failure is a serious chronic medical condition that negatively impacts daily living. Living with heart failure can be challenging due to the physical symptoms, unpredictable nature of the disease, and lifestyle changes required. The objective of this study was to explore and describe the experiences of people living with heart failure and how it affects their health-related quality of life in Ethiopia.

### Methods

A qualitative descriptive design was employed to explore the experience of people living with heart failure, guided by the Theory of Symptom Management. A purposive sample of 14 participants was recruited from the cardiac outpatient clinics at two tertiary-level public hospitals in Ethiopia. Data were collected using a semi-structured interview. The recorded interviews were transcribed verbatim in Amharic, translated into English, and entered NVivo statistical software for analysis. An inductive-deductive hybrid thematic analysis method was used to analyse the data.

### Results

Three themes were identified deductively, while an additional three themes emerged inductively:—"Journey from diagnosis to daily life with heart failure"; "Symptom experience"; "Impact of heart failure on health-related quality of life"; "Perception of health-related quality of life and influencing factors"; "Symptom management and coping strategies"; and "Challenges faced in the journey of living with heart failure". Fatigue, and depression were the most frequently reported symptoms. Participants described how their condition affected their overall physical functioning. Participants utilized consistent follow up-care, adhered to their medications, ensured adequate rest, made dietary modifications, sought social support and engaged in spiritual activities to manage their symptoms and cope with their condition.

**Data Availability Statement:** All relevant data are within the manuscript and its Supporting Information files.

**Funding:** The authors received no specific funding for this work. It is part of a PhD thesis by HM. HM is a higher degree research candidate at UTS, supported by the International Research Training Program (IRTP). The IRTP is a commonwealth scholarship funded by the Australian government and the Department of Education and Training.

**Competing interests:** The authors have declared that no competing interests exist.

**Abbreviations:** DM, Diabetes Miletus; HF, Heart Failure; HRQoL, Health-Related Quality of Life; HTN, Hypertension; IRB, Institutional Review Board; NYHA, New York Heart Association; TSM, Theory of Symptom Management; UK, United Kingdom; UTS, University of Technology Sydney.

Challenges they faced included financial difficulties, unavailability of medications, and a lack of continuity of care.

## Conclusion

People living with heart failure in Ethiopia experience various symptoms. The impact of heart failure on various aspects of their lives, combined with the challenges they face while living with heart failure, significantly affect their health-related quality of life. Health care providers caring for these people need to understand their experiences and the impact on their daily life. Effective multimodal interventions are needed to reduce the impact of heart failure and improve health-related quality of life in this population.

## Background

Heart failure (HF) is a major cardiovascular problem that affects about 2% of the general population in developed countries; however there is little quality HF data from developing countries [1, 2]. It is a significant public health challenge in low-resource settings, including Ethiopia, where it is associated with poor health-related quality of life (HRQoL). In Ethiopia, HF is a serious issue and the most common reason for hospitalisation. It primarily affects middle-aged adults (45 to 65 years old) who are typically the most productive in terms of their roles within family and community [3, 4]. As a result, there is a significant economic impact at an individual and family, as well as national level. HF leads to physical, emotional, and social disruption throughout its disease trajectory [5, 6].

The experience of people with HF failure is significant due to the debilitating and unpredictable nature of the disease. They experience a wide range of symptoms and psychosocial challenges compared to healthy individuals [7, 8]. Common symptoms associated with HF include severe fatigue, dyspnoea, orthopnoea, oedema, chest pain, cough, and palpitations. Additionally, these people frequently experience mental health issues such as depressive symptoms and anxiety [9]. These symptoms, which are responsible for 90% of global hospitalisation among adults with HF, worsen HRQoL by adversely affecting physical, emotional, social, and physiological functioning [8, 10–12].

Treatment options for HF includes pharmacological therapy, symptom management and lifestyle changes, as it is not considered to be a curable condition [13]. People with HF integrate various approaches to cope with their illness [14]. These include taking prescribed medications, embracing spirituality, seeking social support, and engaging in self-management [15–17]. In addition, sharing ideas and experiences with others and being flexible to changing circumstances are ways people with HF cope with their condition [18, 19].

Exploring the symptom experiences of people living with heart failure and understanding how they adapt, along with the coping strategies they employ, is important for comprehending the impact of HF on HRQoL [20]. Addressing symptoms and improving HRQoL are the major aims of HF management [17]. Heart failure is a significant problem in Ethiopia [4, 6, 21] however, little is known about the impact of this condition from the patient perspective. The aim of this study was to explore and describe the experiences of Ethiopian people living with HF, and how it affects their HRQoL, using the Theory of Symptom Management. The findings will provide high-quality evidence for developing interventions to effectively support this population and improve their HRQoL.

## Methods

### Study setting and period

The study was conducted at the cardiac outpatient clinics in two governmental hospitals, St. Paul's Hospital Millennium Medical College and St. Peter Specialised Hospital, in Addis Ababa, the capital of Ethiopia. These settings were selected because they are the largest tertiary-level hospitals, providing cardiology services and treatment for people from all around the country. The hospitals collaborate to provide care for people with cardiac conditions.

The interviews were conducted between 01 February to 31 March 2023.

### Study design

A qualitative descriptive design was used to explore the experiences of people living with HF, following a quantitative survey.

### Theoretical framework

This study was guided by the Theory of Symptom Management (TSM), a middle-range nursing theory to guide research and practice [22, 23]. The TSM is a framework primarily used for nursing research that focuses on three domains (person, environment, and health/illness) and examines the relationship between them. The three essential concepts of the TSM are symptom experience, symptom management strategies, and outcomes (Fig 1). A symptom is a subjective experience reflecting changes in the biopsychosocial functioning, sensations, or cognition of

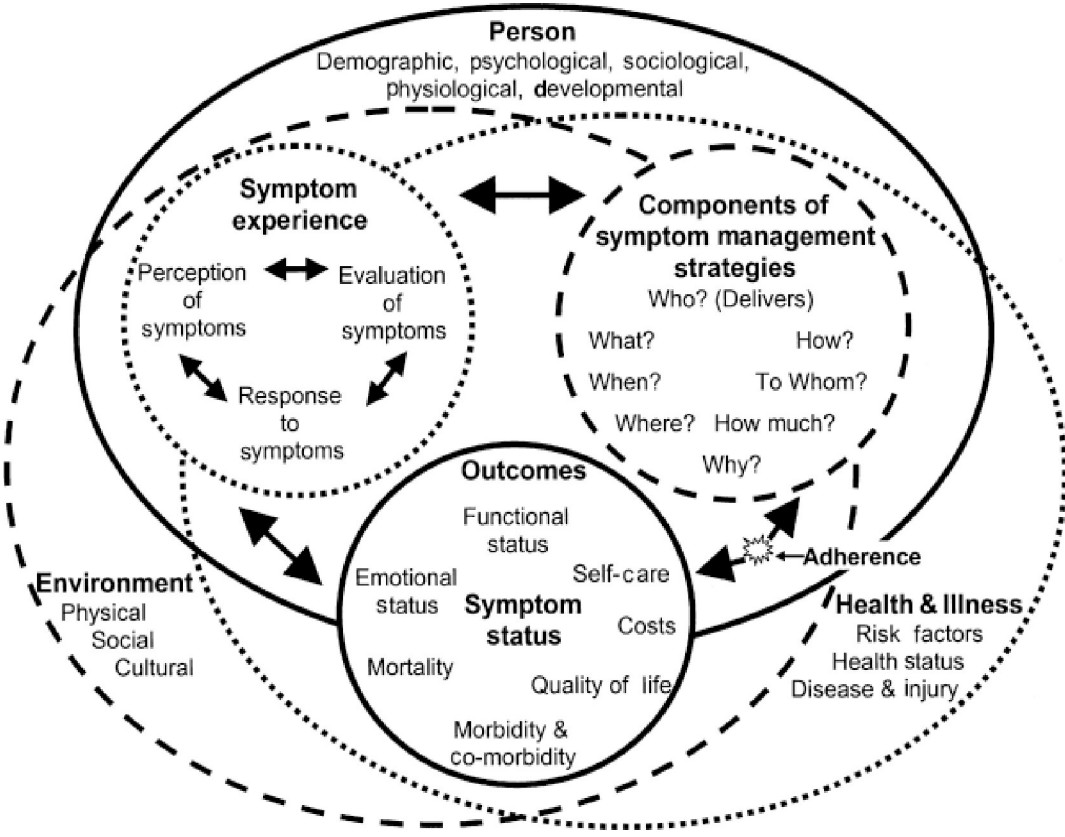

**Fig 1. The revised Theory of Symptom Management [24].**

an individual. Symptom experience is a simultaneous perception of a symptom, evaluation of the meaning of a symptom, and response to the symptom. Symptom management strategies are efforts to avert, delay, or minimise the symptom experience through biomedical, professional, and self-care strategies. Outcomes arise from both symptom management strategies and the symptom experience. These outcomes include obvious improvement in symptom status, which can lead to enhanced physical and mental functioning, and an improved quality of life [12, 14, 17]. The TSM was utilized in this study as it provides a robust framework for understanding the multifaceted experiences of people living with HF, particularly symptom experience, symptom management strategies and impact on HRQoL. This theory was used to formulate the research questions and guide the data analysis.

## Sampling and recruitment

Participants were adults (aged 18 years or older) who had received a clinical diagnosis of HF by a specialist medical doctor and had been receiving regular follow-up care at a cardiac outpatient clinic for at least three months. Invitation letters were distributed to all eligible participants during the earlier phase of the project (quantitative survey) [25] to involve them in the subsequent phase (semi-structured interviews), and if interested, they were asked to provide their phone number to be contacted for a qualitative interviews. Subsequently, the principal researcher (HM) contacted the eligible participants by telephone and invited them to participate in an interview. If they agreed, an appointment was scheduled at their convenience. Participants were recruited using a purposive sampling technique, considering a balanced representation in terms of gender, age, geographical location (urban/rural), and diseases severity. Information power was used to determine the final sample size [26].

## Data collection

The first author (HM), who had undergone a rigorous training and gained experience in conducting qualitative interviews, conducted the interviews using a semi-structured interview guide developed using the TSM [24] and the findings of the quantitative survey published in Scientific Reports [25]. The interviews took place in a private room at the hospital when participants were attending outpatient cardiac clinic appointment. Prior to the interview, brief demographic and clinical information was collected. The audio-recorded interviews were conducted in the local language (Amharic), and notes were taken. The interview guide covered four main topics: sociodemographic and clinical characteristics, symptom experience, concept of HRQoL and impact of HF on it, and symptom management and coping strategies (Table 1). Probing questions were used as needed to explore the impact of participants' symptoms on their HRQoL and how they coped with this. The interviewer transcribed each participant's interview verbatim in Amharic. Signs of information power were observed after 11 interviews. An additional three interviews were conducted to confirm that no significant new information emerged, solidifying the data saturation point. The interviews varied in duration between 30 and 60 minutes.

## Ethical consideration

The study was approved by the University of Technology Sydney (UTS) Human Research Ethics Committees (UTS HREC REF NO. ETH21-6739). Permission to conduct the study was also granted from the institutional review board (IRB) of each hospital (SPHHMC IRB Ref No. PM23/9235/13/10/22, SPSH IRB V561/11/10/22) where the study was conducted. Prior to participation, an information sheet prepared in local language was provided and, if required, read out to all participants. Written consent was obtained from each participant for their interview

**Table 1. Interview guide exploring the experience of people with heart failure in Ethiopia.**

| No. | Primary Question | Optional prompt questions |
|---|---|---|
| 1 | To begin with, I'm going to ask you for some socio-demographic and clinical questions. | *Age, sex, residence, education, marital status, New York Heart Association (NYHA) class, employment status, duration of HF, known comorbidity?* |
| 2 | *Tell me about living with HF?* | *When were you diagnosed and how did you feel at the time?* |
| | | *What kind of symptoms do you have?* |
| | | *What makes you feel better or worse?* |
| | | *What management/treatment strategies do you use?* |
| | | *What about non-medical interventions?* |
| 3 | *Now, I want to discuss the concept of HRQoL.* | *What does the term HRQoL mean to you?* |
| | | *What is the quality of your life since having HF?* |
| | | *Are there things you can't do so well?* |
| | | *What do you do to feel better?* |
| | | *How do you deal with feeling down or depressed?* |
| | | *What are the best and worst aspects of your life now?* |
| 4 | Now, I would like to ask you about the impact of HF on your HRQoL. | *How is it living with heart failure?* |
| | | *How does it affect your family or friends?* |
| | | *What about work? income? sleep?* |
| | | *What about social activities and relationships?* |
| | | *What factors most affect your HRQoL? Why?* |
| | | *What are the most difficult aspects of HF?* |
| | | *How do you deal with these?* |
| | | *What strategies do you use to improve your HRQoL?* |

to be audio-recorded, transcribed, and for the findings to be shared through presentations and publications. Privacy and confidentiality of the participants was maintained throughout the study. Personal details were systematically removed during transcription process.

## Data quality and trustworthiness

Three pretest interviews were conducted before the main interview process. These pretest interviews were used to refine the interview guide, ensuring that the questions were clear and relevant. The data from these interviews were not included in the analyses. To ensure trustworthiness of the data, criteria of credibility, dependability, confirmability, and transferability were applied throughout the research process [27]. Credibility was achieved by allocating adequate time for the interviews and using participants' own words during analysis. Dependability was maintained by keeping thorough records and notes of all the research activities for review when necessary and engaging in debriefings with the second author. To establish confirmability, data were continuously checked and rechecked throughout the study to address any potential biases. Transferability was assured by providing a detailed description of the study population, context, study area, the process of data collection and by maintaining transparency about data analysis.

## Data analysis

Interviews were transcribed verbatim in Amharic and then translated into English by a bilingual expert. One of the authors (HM) actively participated during transcription and translation of the interviews, providing the opportunity to become more familiar with the data. The participants were assigned identification (P01 to P14) to maintain anonymity. The interviews

were imported into NVivo Version 12 statistical software for analysis [28]. An inductive-deductive hybrid thematic analysis approach was utilised to analyse the data. This approach was chosen because the research questions were strongly linked to the existing theory (deductive), and it enabled the exploration of insights within the data while allowing new themes to emerge from the data (inductive). The validity and rigour of this approach has been well demonstrated previously [29–31]. The overall analysis was undertaken across six phases following Braun and Clark's thematic analysis method: data familiarisation, systematic data coding, searching for themes deductively using TSM, creating new themes inductively, reviewing themes, defining, and naming themes, and producing the final report [32]. Before the analysis, the authors read and re-read the transcribed/translated data to immerse themselves and become familiar with the concepts for a holistic understanding of the data. During data familiarisation, memos and annotation were taken to document important findings and insights. After data familiarisation, both semantic and latent coding were utilised to generate codes for significant information from each transcript. To enhance the validity of coding, two authors (HM and PS) independently coded the transcripts, and the third author (AW) was consulted as needed to resolve any coding discrepancies. As coding progressed, similar or related codes were grouped into categories, and these categories were used to identify and generate themes and subthemes. The final identified and emerged themes were critically discussed and approved by all authors.

## Results

### Sociodemographic and clinical characteristics of the study participants

A total of 14 people (eight male and six female) with HF were recruited and completed the study. The age range of the participants was 32 to 73 years, with a mean age of 52.92 years. Over half (57.14%) of participants were married, and half (50.00%) had completed primary education. Half (50.00%) were employed, 21.43% were retired, and 28.57% were unemployed. Duration of HF varied with 28.6% of participants having lived with the condition for more than five years. The New York Heart Association (NYHA) classification was used to determine the severity of HF, revealing 43.00% of participants classified as class III or IV. Details of sociodemographic and clinical characteristics are summarised in Table 2.

### The experience of people living with heart failure

The inductive-deductive hybrid thematic analysis method revealed three themes (Theme 2, 3 and 5) that were identified deductively, while an additional three themes (Theme 1, 4, and 6) emerging inductively, all containing related subthemes (Table 3).

 While these six themes are presented separately, they were all inextricably linked. Analysing the relationship between these themes provides insights into the holistic experience of people living with heart failure in Ethiopia (Fig 2).

### Theme 1: "Journey from diagnosis to daily life with HF"

This theme encompasses participants' initial feelings and thoughts following their life-altering diagnosis. Participants described their experiences after receiving the diagnosis and their progress after medical care was initiated.

 *Subtheme 1.1*: *"Initial reaction to diagnosis"*. Participants reported feeling shocked, anxious, and hopeless upon receiving their diagnosis.

 *"I was shocked to hear the diagnosis and thought I wouldn't be able to survive." (P05)*

**Table 2. Sociodemographic and clinical characteristics of participants.**

| Participants | Age | Sex | Marital status | Employment status | Education | Duration of HF | Known comorbidity | NYHA class |
|---|---|---|---|---|---|---|---|---|
| P1 | 71 | M | Married | Unemployed | Primary | 6 years | DM | II |
| P2 | 73 | M | Widowed | Retired | Secondary | 3 years | No | III |
| P3 | 52 | M | Married | Employed | Degree | 1 year | DM | I |
| P4 | 60 | M | Married | Employed | Primary | 11 years | Asthma | IV |
| P5 | 32 | F | Single | Employed | Primary | 3 years | No | II |
| P6 | 48 | F | Married | Unemployed | Primary | 8 years | No | III |
| P7 | 62 | M | Married | Retired | Degree | 1 year | HTN, DM | II |
| P8 | 40 | M | Married | Employed | Primary | 1 year | No | III |
| P9 | 65 | F | Widowed | Unemployed | Primary | 2 years | No | I |
| P10 | 70 | F | Widowed | Retired | Primary | 4 years | DM, HTN | IV |
| P11 | 35 | F | Single | Employed | Diploma | 3 years | No | I |
| P12 | 42 | F | Married | Employed | Secondary | 8 years | No | I |
| P13 | 43 | M | Single | Unemployed | Diploma | 2 years | Epilepsy | III |
| P14 | 48 | M | Married | Employed | Secondary | 1 year | HTN | II |

DM: Diabetes mellitus, HTN: Hypertension, NYHA: New York Heart Association

*"After I learned about my condition, I was anxious. I couldn't sleep at night due to my anxiety. I thought I would die." (P12)*

Some participants voiced concerns about their children's future, while others expressed a feeling that their condition was part of a divine plan.

*"I was shocked when I heard of my diagnosis since I have two kids and felt like how they would survive if I'm leaving them; I was worried about them, not about myself". (P14)*

*"I wasn't shocked to hear of my diagnosis, it happened as per God's will". (P10)*

**Table 3. Themes and subthemes in the study on experiences of people with HF in Ethiopia.**

| No | Theme title | Subtheme | Origin |
|---|---|---|---|
| 1 | Journey from diagnosis to daily life with HF | • Initial reaction to diagnosis<br>• Daily life in post-HF diagnosis<br>• Progress after medical care | Emerged |
| 2 | Symptom experience | • Physical symptoms<br>• Emotional symptoms<br>• Psychosocial symptoms | Identified |
| 3 | Impact of HF on HRQoL | • Physical impact<br>• Emotional impact<br>• Social impact<br>• Sexuality impact | Identified |
| 4 | Perception of HRQoL and influencing factors | • Perception of HRQoL<br>• Influential factors | Emerged |
| 5 | Symptom management and coping strategies | • Self-management<br>• Behavioural management<br>• Social support<br>• Spiritual activities | Identified |
| 6 | Challenges faced in the journey of living with HF | • Financial difficulties<br>• Availability and high cost of medications<br>• Lack of continuity of care | Emerged |

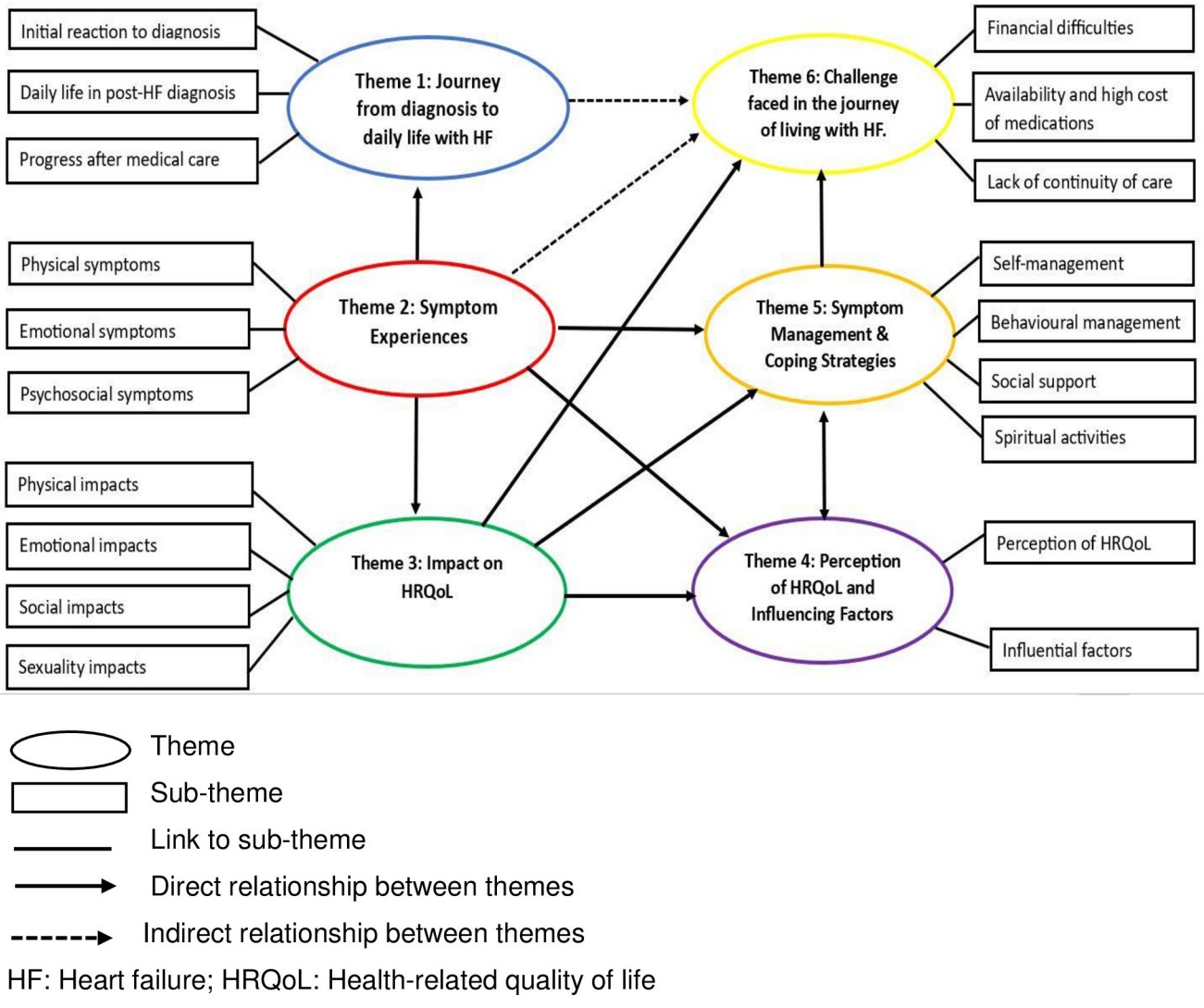

**Fig 2. Thematic map indicating the relationship between the six themes.**

*Subtheme 1.2*: *"Daily life in post-HF diagnosis"*. Most of the participants reported that their lives had changed significantly post diagnosis, especially compared to their pre-diagnosis lives.

*"My lifestyle is not as it was before. You should deal with it. I was doing all sorts of activities without difficulty...you know what...I did everything at home, but now I have limitations, a heavy activity makes me to feel tired" (P11)*

*Subtheme 1.3*: *"Progress after medical care"*. Many of the participants reported noticing positive changes post-medication and after follow-up care. Adherence to medications and the self-care recommendations had contributed significantly to their overall good progress.

*"I thought I wouldn't improve at all. But I have seen positive changes in my health status after I started taking medication and received follow-up care in this hospital." (P07)*

## Theme 2: "Symptom experience"

Participants described their symptoms, including physical, emotional, and psychosocial symptoms. Every participant reported having at least one symptom, with physical symptoms being the most frequently mentioned.

*Subtheme 2.1*: *"Physical symptoms"*. Many participants mentioned several physical symptoms, with fatigue and exhaustion being the most frequently cited symptoms.

*"Currently, I have chest pain, fatigue, dyspnoea, abdominal pain, leg swelling, and decreased appetite. I have difficulty breathing and I can't walk for long distances." (P04)*

*Subtheme 2.2*: *"Emotional symptoms"*. Participants often described experiencing depressed or anxious feelings as a result of the diagnosis and attributed them to physical limitations and lifestyle changes post-diagnosis.

*"I feel down because I can't work as hard as I used to. . . I become fatigued after working hard. . .you know. . . in that case I feel depressed. I asked God . . . why is this happening to me?" (P05)*

*Subtheme 2.3*: *"Psychosocial symptoms"*. Participants expressed experiencing a range of psychosocial symptoms including sadness, frustration, isolation, worry, and sudden mood changes.

The feeling of worry was commonly described due to the unpredictable nature of the disease, which made them feel unsure about what might happen next.

*"I am scared of falling asleep because I feel like I'm going to die, and I worry a lot that one day my heart may stop beating." (P12).*

Some participants experienced sudden mood changes upon realising the impact of their condition on daily functioning.

*"Your mood suddenly changes when you realize that you have physical limitations. I usually want to be alone. . .I require silence; I don't want to be disturbed by any noise, not even by the voices of my children." (P06)*

## Theme 3: "Impact of HF on HRQoL"

Participants reported how HF had impacted their everyday lives. They described a multifaceted experience, and four subthemes emerged from the data: physical impact, emotional impact, socio-economic impact, and impact on sexual relationships.

*Subtheme 3.1*: *"Physical impact"*. Many of the participants described how their condition affected their daily activities, work, and overall physical functioning. Activities that were simple and taken for granted were now difficult for them.

*"I can't walk far, and I can't finish what I start. . . . it is even hard for me to come from home to this hospital for my regular follow-up. You can't do what others are doing; you have limitations. When you go to do something, and you realize that you can't do it. For instance, one day I was walking fast to take the bus. I saw 70-year-old women pass me for the bus, but I was late to reach the bus, and it left. At that moment, I really felt like I was worthless. I became panicked, wondering why this was happening to me." (P08)*

Similarly, Participant 11, a 35-year-old female, reiterated: *"I can't do things like I used to. I have difficulty of walking long distances at a fast pace. I find it challenging to go shopping because I can only lift and carry up to five kilograms."*

*Subtheme 3.2*: *"Emotional impact"*. Many participants expressed feeling negative emotions including depression and anxiety as a result of the physical impact of their condition.

*"I feel down when I get tired of doing ordinary activities at home. . .when I feel depressed, I have nothing to do. . .just nothing, I just sit down and take a rest and there is no other solution. I can't eat what I want. When I see someone else eating something delicious, I become frustrated because I can't. My life right now is very miserable. I'm angry that I can't go out and come back as I want. This is the worst. I might seem in good physical condition, but my body is weak, which depresses me."* (P04)

*Subtheme 3.3*: *"Social impact"*. Many participants indicated that HF led to social isolation, and challenges in maintaining relationships.

*"I used to actively engage in social activities, visit my relatives on holidays, visit sick people, and have fun with my friends. But now, due to my illness, I stopped everything. . . I mean, I can't do that now. People may also isolate you; for example, when you cough, they may treat you differently and think you have COVID 19. You find this upsetting."* (P07)

A 52-year-old male, shared a workplace experience in which he faced challenges with his colleagues:

*"At work, my co-workers are more frustrated with my condition than I am. They feel that people with heart failure are more vulnerable to sudden death. For instance, if you need to have a loan of money, nobody wants to take the risk by giving it to you because they worry that you might die suddenly before paying them back."* (P03)

*Subtheme 3.4*: *"Sexuality impact"*. Two participants indicated that they had experienced reduced sexual desire and the inability to perform sexual intercourse with their partners. Both participants were concerned about whether it was due to the disease process itself or the side effects of medications.

*"This illness has also had an impact on my marriage because I feel less motivated to engage in sexual activities. I used to have a good sexual relationship with my wife, but now it has impaired my entire sexual desire. My wife thinks that I am having a relationship with another woman because of my lack of interest in intimacy with her. I'm not sure if this is due to the side effects of the medications or something else."* (P08)

### Theme 4: "Perception of HRQoL and influencing factors"

Participants described their perception of their HRQoL and evaluated their overall well-being in the context of their physical health, mental health, and social functioning. This perception varied from participant to participant and was influenced by various factors.

*Subtheme 4.1*: *"Perception of HRQoL"*. Some participants perceive HRQoL primarily in terms of emotional well-being, happiness, and physical activity; they feel that feeling content, balanced emotionally, and engaging in regular physical activity are essential aspects of living a

life of good quality. Others, however, focus more on financial aspects when considering HRQoL; they feel that financial stability leads to a sense of security and facilitates access to resources that can improve overall well-being.

> *"Health-related quality of life means the physical ability to do what you want and being physically healthy, no illness." (P03)*

> *"I think staying healthy and financially secure are key components of health-related quality of life. If I have adequate income, I could be happy and comfortable with my life. . .. I think that is QoL." (P07)*

*Subtheme 4.2*: *"Influential factors".* Many participants described the influence of several factors on their HRQoL including age, symptoms, comorbid conditions, income, social support, and general health perception.

Participants perceived that life slows down as age increases, which impacts their overall health status.

> *"Everything slows down as you age; the presence of heart failure exacerbates this and has a substantial impact on my day-to-day activities, which influenced my quality of life." (P03)*

Physical and emotional symptoms, such as chest pain and depression, had a negative effect on HRQoL.

> *"The presence of symptoms affects your quality of life since you can't work even though you want to, and when you try to work, it makes you feel exhausted, incapacitated, and possibly weaker than you once were so I sit down. Yes, obviously, it has a significant influence." (P06)*

Participants perceived that the presence of comorbid conditions also had a negative impact on their physical and mental health.

> *"I have additional diabetes and high blood pressure. . .so I am scared. . .what is going to happen? You become dissatisfied with your life when you have many comorbid conditions." (P10)*

Some participants had lost their job due to their physical limitations, which reduced their income. Unemployment and financial hardship led to anxiety and depression.

> *"My condition prevents me from working as I did before, so I reduced on my working hours. When I start working, you know. . .. I feel fatigue and tired. I can't work the whole day, so I get frustrated with not being able to work and not having the energy as before. So, I am not working as before because of my illness, and it affects my income. In this case, you feel dissatisfied with your life and feel down. "(P14)*

Participants highlighted that living without social support was quite challenging with some emphasising the need for someone to provide physical and/or financial support while living with HF.

> *"Social support has an impact on my quality of life because when I'm away from my family, I'm anxious, but when they're nearby, I feel relaxed and safe. They support me in all my home activities, and without them I would not be alive at the time." (P12)*

Many of the participants perceived themselves as being in poor health status, which contributed to feelings of depressive symptoms, thereby reducing their HRQoL.

*". . .. here I am, as you can see, a critically ill and weak person. I asked God if my time is over; I don't want to suffer anymore, and I am not scared of dying. The fatigue is overwhelming; I can't even do simple tasks without difficulty. It's been really challenging. I am exhausted with everything, and I feel like my health is deteriorating. I am not a happy person now because I am not feeling well. I never thought I would have to go through all this. . ." (P10)*

### Theme 5: "Symptom management and coping strategies"

Participants reported strategies they use to manage their symptoms and cope with their condition. This theme includes four subthemes: self-management, behavioural management, social support, and spiritual activities.

*Subtheme 5.1*: *"Self-management"*. Several self-management strategies were reported including medication adherence, getting adequate rest, attending regular follow-up care, making dietary modifications, engaging in exercise and weight monitoring, use of herbal products, and avoiding harmful habits.

*"When I feel tired, I sit down and take a rest. This makes me to feel better." (P03)*

"I monitor what I eat; I avoid salt; and I am using a sunflower oil instead of eating fatty foods. Of course, I reduced salt and avoid fatty foods.". (P07)

*"I'm taking my medicines properly as prescribed. After I started taking the medicines, my symptoms have improved." (P09)*

*Subtheme 5.2*: *"Behavioural management"*. Behavioural strategies included acceptance of condition, distraction, maintaining a positive emotional state, and avoiding stress. Participants indicated that accepting their condition helped them cope with it.

*"It is a matter of accepting your condition. If you do not accept it, you will be depressed. Although I have heart failure, this does not necessarily mean that I am going to die. I accepted my condition and am following the instructions from the doctors." (P07)*

Some participants reduced their symptoms and managed their condition by entertaining themselves using various techniques.

*"I have always tried to feel happy, and I don't want to feel down. I watch television, I play and talk with my family members so that I can forget about my depression." (P01)*

*Subtheme 5.3*. *"Social support"*. Many participants mentioned the importance of social support from family and others to cope with this condition. This view emphasizes how important social support may be for people with chronic conditions. Social support from friends, family, and loved ones can provide a sense of belonging, and physical, and emotional support—all of which are essential for navigating challenges associated with HF and improving HRQoL.

*". . . my husband and children are helping me a lot, and I feel comfortable. My brothers who live abroad also provide me with support, assisting me in managing my health and sometimes sending me money and medications. Their support has been crucial to my survival, and I would find it difficult to live without them." (P06)*

Participants indicated that sharing experiences with other patients helped them to cope with it.

*"I shared ideas with other people while I came here for my appointment. We shared strategies for dealing with our symptoms and other problems, which helped me to reduce my depression and made me feel good." (P12)*

*Subtheme 5.4*: *"Spiritual activities"*. Several participants emphasised the importance of spirituality in coping with the emotional and physical challenges of living with HF. Attending church regularly and praying to God were part of their day-to-day activities.

*"I am coping with the help of my God. I feel I have a good relationship with God. I am always praying and reading the Bible. I might not be able to do things I used to do, but I believe God is with me. . . I am not afraid of dying." (P13)*

## Theme 6: "Challenges faced in the journey of living with heart failure"

Participants mentioned a wide range of challenges they faced during their journey of living with HF including financial difficulties, the availability of medications and their high costs, as well as a lack of continuity of care.

*Subtheme 6.1*: *"Financial difficulties"*. The high cost of medical care and medications posed a significant financial strain for almost all participants. They reported that medical expenses were hard to afford, and for those who could not afford, it felt like a death sentence.

*". . . . . . it's a death sentence if you have a cardiac condition, don't have enough money, or can't afford it." (P03)*

*"The expenses related to managing heart failure are high, and there are several costs involved, including those for medications, blood tests, and diagnostic procedures. Thus, if you can't afford them, you'll feel miserable, your symptoms will get worse, and you might even die." (P12)*

*Subtheme 6.2*: *"Availability and high costs of medications"*. Many participants reported that unavailability and the high costs of prescription medicines were extremely challenging.

*"Purchasing medicines on the open market can be challenging and expensive. I buy what I need if it is within my budget; however, if it is not affordable, I will not purchase that medication. In such instances, not having the medication makes my symptoms worse." (P08)*

*Subtheme 6.3*: *"Lack of continuity of care"*. Some participants expressed their frustration regarding the lack of continuity of care during their monthly hospital visits. They indicated that inconsistent physician involvement and inadequate communication between healthcare providers were significant concerns.

*". . .. when I arrived for my appointment, the physicians occasionally changed. I never saw the same physician twice. Due to the doctors' poor handwriting, I have noticed them getting confused about my most recent medical history when reading my chart. Their colleagues' handwriting was also difficult for them to read. Consequently, they just continued prescribing the same medication. They responded to my concerns quickly, didn't fully consider my*

*complaints, and seemed eager to rush through the appointment to move on to the next waiting patient." (P02)*

*"Since starting follow-up care in this hospital, I've started taking medications and have regular appointments every month. However, no doctor has consistently examined me. I expect doctors to provide adequate information about my progress, but they often ask questions quickly and then hand you a prescription. I once provided a blood sample, but they haven't returned the results. I continue to come here simply to take my medications." (P08)*

## Discussion

Heart failure is a serious chronic medical condition that has an adverse impact on HRQoL [17]. To our knowledge, there needs to be more data regarding the experience of people with HF in Ethiopia and how it affects their HRQoL. A recently published study conducted in Northwest Ethiopia by Mengistu et al. (2024), during the same data collection period as our study but with a two-month gap, revealed the physical, social and emotional impacts of HF, and the challenges associated with HF treatment [33]. Expanding upon these findings, the current study delves deeper into these issues, providing additional insights regarding symptoms experience, symptom management and coping strategies, and perception of HRQoL and influencing factors in this population. These additional insights are critical for a comprehensive understanding of the phenomenon.

In this study, six main themes were identified and emerged from the data. Under "Journey from diagnosis to daily life with HF," participants described initial shock and anxiety post-diagnosis, followed by significant lifestyle changes and a gradual adaptation to living with HF. The "Symptom experience" theme revealed struggles with physical, emotional, and psychosocial symptoms, such as fatigue, chest pain, depressive symptoms, and anxiety. "Impact of HF on HRQoL" encompassed the extensive physical limitations, emotional distress, social challenges, and disruptions in sexuality. In "Perception of HRQoL and influencing factors," participants' views on quality of life were shaped by age, symptoms, comorbidities, income, health perception and social support. The "Symptom management and coping strategies" theme highlighted approaches such as medication adherence, lifestyle modifications, and seeking social and spiritual support. Lastly, "Challenges faced in the journey of living with HF" underscored financial difficulties, medication accessibility issues, and inconsistencies in healthcare. Overall, this study provides a holistic view of the complex and multifaceted impact of HF on individuals in Ethiopia.

The findings emphasise the significant psychological distress accompanying an HF diagnosis, echoing results from UK-based research [34]. Participants commonly experienced shock and sadness, stemming from fears about mortality and potential physical limitations [35, 36]. This emotional impact might be exacerbated by a lack of prior HF knowledge [37], highlighting the need for improved education and early intervention strategies. Consistent with other studies [38–40], the diagnosis necessitated lifestyle changes and presented new challenges, reinforcing the notion that HF affects multiple life dimensions [41, 42]. These findings further highlight the need to develop patient education programs to help people understand their condition, manage symptoms, and adapt to new lifestyle requirements effectively.

The categorisation of symptoms into physical, emotional, and psychosocial domains provides a comprehensive understanding of the participants' health challenges. Commonly reported physical symptoms, such as fatigue, tiredness, shortness of breathing, and chest pain, align with previous findings [12, 43, 44]. Predominantly, fatigue affects the daily activities of participants and causes frustration. This aligns with existing literature [45–47], and

underscores the need for targeted management strategies. The association between fatigue and emotional distress [41], particularly depressive symptoms [47], shows the importance of holistic care addressing both physical and emotional health. Emotional symptoms align with prior studies [48–50], highlighting the impact of financial difficulties, uncertainty about their condition, and physical limitations as major contributors to depressive symptoms [35]. This emphasises the need to consider not only physical, but also financial and psychosocial aspects in the management of people with HF.

Participants echoed concerns found in previous studies about the restriction of daily activities and social participation due to HF [44, 51, 52]. These limitations, coupled with fatigue and weakness [53], not only diminish physical capacity but also led to social isolation [51] and emotional distress [34, 54–56]. The impact of HF on sexual intimacy, as noted by some participants, highlights the broad-ranging effects of the condition on personal relationships [57]. Two participants described how their condition impacted their sexual intimacy with their partners. This might be due to physical limitations, medication side effects, and emotional factors [57]. To address these issues, interventions should focus on tailored exercise programs to improve physical functioning and provide emotional support for adapting to lifestyle changes.

Perceptions of HRQoL, consistent with definitions found in previous research [46, 58], focusing on happiness, income, and health, diverge from standard definitions, potentially due to lower educational levels affecting comprehension [55]. This finding suggests a need for healthcare providers to contextualise HRQoL in appropriate terms for people with varied educational backgrounds. The influence of age, comorbid conditions, depression, income, health perception and social support on HRQoL [59–62] indicates that a multifaceted approach that accounts for the diverse factors affecting HRQoL is essential in managing HF.

Participants utilised multiple strategies for symptom management and coping. This study aligns with previous research [50, 55, 56] in highlighting the prevalence of self-management strategies like medication adherence and lifestyle modifications. The importance of behavioural strategies, including acceptance and seeking social support [12, 19, 50, 55], is evident. These findings reinforce the role of comprehensive patient education and support systems in enhancing coping mechanisms and self-care. Social connections play a pivotal role, with participants stressing the positive impact of social support [12, 34, 58] and shared experiences [63, 64] on coping abilities, and the need for holistic care that addresses psychosocial well-being. Additionally, spiritual activities, which have also been reported in the previous research [12, 65, 66], were identified as significant coping mechanisms, highlighting the potential benefits of integrating spiritual support into HF care plans.

Financial constraints were a significant challenge for participants, with concerns about out-of-pocket healthcare costs highlighting the broader socioeconomic factors influencing HF management. The unavailability of medication in the open market, coupled with financial difficulties, was associated with poor outcomes, such as depression and poor HRQoL [67, 68]. These challenges are probably due to insufficient health care financing, and lack of adequate insurance coverage [69, 70]. This emphasises the necessity of policy interventions, including the enhancement of healthcare infrastructure, improvement of medication supply chains, and provision of financial support for HF patients to effectively address the identified challenges. Participants expressed frustration at the lack of continuity of care and poor interprofessional communication. This could be because most of the doctors in the study hospitals' cardiac clinics were resident physicians who rotate on a monthly basis. This issue, consistent with findings in other studies [38, 54, 71], illustrates the importance of consistent and effective care for individuals with chronic conditions. The reported poor communication with healthcare providers [39, 72] shows the importance of establishing strong patient-provider relationships to improve outcomes and HRQoL.

This study provides insight into the multifaceted impact of HF in Ethiopia, from initial diagnosis to ongoing management. It highlights the need for holistic care approaches that address physical, emotional, and social needs, and the importance of patient education, consistent healthcare provision, and support systems in improving the HRQoL for individuals living with HF. The findings have significant policy implications, particularly in resource-limited settings. Further research to understand cultural and contextual influences on HF management, particularly in settings like Ethiopia could help in developing culturally sensitive and contextually appropriate intervention strategies.

## Limitations

There are some limitations that should be considered for future research. Potential for selection bias may arise as people who chose to participate in the study might have different experiences or views compared to non-participants. The identification and interpretation of themes may be influenced by the researchers' perspectives, potentially leading to biased conclusions. Cultural and geographical characteristics of the sample may limit the transferability of findings to other contexts, reducing generalisability. Furthermore, language and interpretation issues may risk losing nuanced meanings in translation.

## Conclusion

This study provides in-depth descriptions of the experiences of people living with HF. The diverse symptoms experienced by participants and their impacts on various aspects of life highlight the profound impact of HF on HRQoL, and the pervasive influence of this condition on individuals' lives. Notably, financial difficulties and medication unavailability emerged as significant challenges, significantly affecting HRQoL. Symptom management and coping strategies such as medication adherence, self-care, spirituality, consistent follow-up care, and social support, underscore the multifaceted approach individuals employ to cope with HF. In response to these findings, there is a clear call for effective multimodal and context specific interventions, such as psychosocial, educational, and lifestyle changes, to mitigate the multifaceted impacts of HF and improve HRQoL in this population.

## Supporting information

**S1 File. Data set.**
(PDF)

## Acknowledgments

We would like to thank UTS and the dedicated staff members of St. Paul Hospital Millennium Medical College and St. Peter Specialised Hospital for their invaluable cooperation and support during the ethical approval process for this study. We would also like to express our gratitude to the respected respondents who participated in this study.

## Author Contributions

**Conceptualization:** Henok Mulugeta.

**Data curation:** Henok Mulugeta.

**Formal analysis:** Henok Mulugeta, Peter M. Sinclair, Amanda Wilson.

**Funding acquisition:** Henok Mulugeta.

**Investigation:** Henok Mulugeta, Peter M. Sinclair, Amanda Wilson.

**Methodology:** Henok Mulugeta, Peter M. Sinclair, Amanda Wilson.

**Project administration:** Henok Mulugeta, Peter M. Sinclair, Amanda Wilson.

**Resources:** Henok Mulugeta, Peter M. Sinclair, Amanda Wilson.

**Software:** Henok Mulugeta, Amanda Wilson.

**Supervision:** Peter M. Sinclair, Amanda Wilson.

**Validation:** Peter M. Sinclair, Amanda Wilson.

**Writing – original draft:** Henok Mulugeta.

**Writing – review & editing:** Henok Mulugeta, Peter M. Sinclair, Amanda Wilson.

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
