## [Decision Letter · Decision Letter 0]

9 Jul 2024

PONE-D-24-12608The experience of people living with heart failure in Ethiopia: a qualitative studyPLOS ONE

Dear Dr. Mulugeta,

Thank you for submitting your manuscript to PLOS ONE. After careful consideration, we feel that it has merit but does not fully meet PLOS ONE’s publication criteria as it currently stands. Therefore, we invite you to submit a revised version of the manuscript that addresses the points raised during the review process.

 Please address all the questions raised by the reviewers. 

We look forward to receiving your revised manuscript.

Kind regards,

Asres Bedaso Tilahune

Academic Editor

PLOS ONE

“This study was funded by University of Technology Sydney (UTS). It is part of a PhD thesis by HM. HM is a higher degree research candidate at UTS, funded by the International Research Training Program (IRTP). The IRTP is a commonwealth scholarship funded by the Australian government and the Department of Education and Training.”

Additional Editor Comments:

Address all the questions raised by both reviewers.

Reviewers' comments:

Reviewer's Responses to Questions

**Comments to the Author**

1. Is the manuscript technically sound, and do the data support the conclusions?

Reviewer #1: Partly

Reviewer #2: Yes

2. Has the statistical analysis been performed appropriately and rigorously? 

Reviewer #1: N/A

Reviewer #2: N/A

3. Have the authors made all data underlying the findings in their manuscript fully available?

Reviewer #1: Yes

Reviewer #2: Yes

4. Is the manuscript presented in an intelligible fashion and written in standard English?

Reviewer #1: Yes

Reviewer #2: Yes

5. Review Comments to the Author

Reviewer #1: Comments to Authors

I believe the manuscript is an important study that contributes to the existing evidence on the experience of people living with heart failure and its impact on quality of life. However, some points need to be elaborated further by the authors to ensure the paper's clarity.

# I would suggest the title need to be modified ‘Experience of people living with heart failure in Ethiopia: a qualitative study’

#Abstract

*Background

I would suggest using full wordage for heart failure in line 33, page 2, as abbreviations are not recommended in an abstract, as all abbreviations should be defined in the background section. Therefore, it is best to remove all abbreviations in brackets in the abstract and only use full wordage.

Line 37, page 2 you can write as ‘A descriptive qualitative ….’

* Result

I would suggest that to increase the transparency and attractiveness of the manuscript. In the results report, the number and list of categories, themes, and sub-themes should be indicated in both the abstract and the text. And the authors should also enclose the themes and subthemes in the quotation marks across the manuscript.

* Line 53, page 2, “People living with HF in Ethiopia experience various symptoms.” What are various symptoms? Based on the authors manuscript findings, the above sentence doesn’t quite inline.

* Line 53-55, page ‘The impact of HF on various aspects of their lives, combined with poor general health perceptions, significantly affects their HRQoL.’ I could not find any mentions ‘…. poor general health perceptions in your findings. Could you clarify whether it is a health care providers or patients who have these perceptions.

* Line 55, page 2, ‘They’ represents ……

*Line 56 and 57, page 2, I could not find any mentions ‘……to understand their experiences and impact on their daily life’ in your findings.

* …what are ‘Effective multimodal interventions….’

I suggest the authors modifying conclusion based on their pertinent findings.

#Keyword, use Mesh terms

#Background

*In the first sentence you explained the prevalence in developed, whereas in the same sentence you discussed quality data in developing countries. It needs to be rewritten.

* Line 66, page 3, ……reason for hospitalisation. It needs to be cited.

* Line 67, page 3, …… within family and community. It needs to be cited.

* Line 71, page 3, …… unpredictable nature of the disease. It needs to be cited. Line 74, page 3, …… and palpitations. It needs to be cited.

* Line 75, page 3, …… as depression and anxiety. It needs to be cited.

*Line 76, page 3, I would suggest using full wordage for HRQoL, the first time it appears in the background section.

*The fourth paragraph needs to be merged with the second paragraph, as both paragraphs explain HF symptoms and HRQoL.

*Added a little bit HF and HRQoL low resource settings and Ethiopia.

*Line 88, page 3…….in Ethiopia (18, 19). Both references do not explain the HF is a significant problem in Ethiopia. what study gaps do authors want to uncover?

#Methods

* Sampling and recruitment

*Who distributed invitation letters to all eligible participants?

*How to compensate participants for their time when the authors invite them to participate the study?

*Participants were recruited using a purposive sampling technique. according to what criteria?

*How did you employ this purposive sampling technique? with who? By whom?

*…. interview guide developed using the TSM. Who was the team that developed it?

*Signs of data saturation were observed after 11 interviews. You used descriptive qualitative approach and used ‘data saturation’ rather than ‘information power’ for sample size determination. ‘Data saturation’ is recommended for grounded theory, but ‘information power’ is recommended for descriptive qualitative approach. Therefore, you should a little bit justify it.

Q#4 is a leading question

*The interviews were imported into NVivo version 12 ‘statistical software’ for analysis?

*Creating new themes inductively, reviewing themes, defining, and naming themes, and producing the final report (31). Use the updated reference and phase.

# Results

*Were participants given a chance to review their transcripts/ summary of findings?

*Line 207, page 8, it is not clear to me ‘Experiences of people living with heart failure’.

*Authors need to write themes and subthemes as word enclose quotation mark rather than number.

*Themes are broadening idea some are overlapped It can be merged.

*Perception of HRQoL and its influencing factors … needs revision.

*It is important for ‘Initial reaction to diagnosis’ subthemes is resynthesis and see quotes.

*Line 232, page 10, …. Most of the participants. There is one quotation, how do say most of the participants?

*The same issue …. line 248, page 10… Many participants…….

*Similar ideas and quotes appear in multiple places. For example, participants reported feeling shocked, anxious in the ‘Emotional symptoms’, ‘Initial reaction to diagnosis’, ‘Emotional impact’ and ‘Factors influencing HRQoL’ subthemes.

*Line 446, page 17, ‘Lack of continuity of care’. It needs revision

*Routine mental health evaluations and counselling as part of HF management could be beneficial. How do you come up in this conclusion?

Reviewer #2: The authors address very important public health concern and come up with good results. For the improvement of the manuscript I forwarded comments in section by section.

Manuscript title

Line 1: ‘’The experience of people living with heart failure in Ethiopia: a qualitative study’’

The experience of people living with HF…..’’experience’’ is broad term better to indicate patient ‘acceptance of their diagnosis, experiencing symptoms, impact of their quality of life and coping strategy’ since the central theme of the manuscript involves in these concepts. Hence, I strongly recommend the revision of the title by considering such concepts.

Abstract

Background: (line 35-36) The objective of this study was to explore and describe the experiences of people living with HF and how it affects their health related quality of life (HRQoL) in Ethiopia. But the title did not indicate the HRQoL, better to consider the title revision above

Methods: line 37-43: better to indicate the theoretical framework you use in the method section of the abstract.

Results: line 44: You mentioned three themes were identified and three were emerged but what are these them should indicated just immediately to the description. This will give clear idea about the themes to the reader.

Conclusion: …. poor general health perceptions’’ but there is not result indicated poor general health perception the results section of this abstract.

Keywords: ‘’experience’’ better to qualify experience associated with my comment in the title revision

Background

Page 3: line 84: the last paragraph first sentence; ‘’exploring the symptom experiences of people living with heart failure and understanding how they adapt’’ better to your title for this study

Line 90-91: ‘’theory of symptom management’’ it is good to indicate why you use this particular theory and give some details just to inform the reader about it. You may use theories either to formulate your study question or do your analysis or both but it is not clear why you use this aprticular theory. Please make it clear. I understand you did not fully using its constructs in either case. Any justification please? The question also extend to the method section too,

Methods

Page 5: Table 1: data collection guide: ‘’Demographics’’ but the contents in the right side are not only demographics rather clinical issues,

Ethical consideration: page 6: line 52-54 ….Permission to conduct the study was also 153 granted from the institutional review board (IRB) of each hospital where the study was 154 conducted. But do these hospitals have IRB? If so better to present the ethical approval number in the similar way done above

Line 161: ‘’Three pretest interviews’’ did you include or exclude in the main analysis? Indicate what do you with these pretest interviews and explain why. Because this is important for the reader and methods section of the qualitative study.

Results

Page 8: ‘’The age range of the participants was 32 to 73 years, with a mean age of 52.92 years.’’ You use range and mean but not correct statics i.e range with interquartile range while mean is ideal for normal distribution with standard error

Table 2: the second colomun in the right end ‘’comorbidity’’ better to designate as ‘’known comorbidity’’ since there may be undiagnosed one

Table 3: you mix-up themes which were pre-identified and emerged. Do you have any reason not to use in the ordered fashion?

First theme ‘’ Experience from diagnosis to daily life with HF’’ and its subthemes are not inline. Subthemes state about people’s acceptance and living but experience is very wider term, I strongly suggest to reconsider the word ‘’experience’’ see my concern in the title too

Last theme:’’ Challenges faced in the journey of living with HF’’ and last subtheme ‘’ Lack of continuity of care’’ does not used in the right way. Continuity of care has different meaning in the health system concept but in your study it was change of healthcare personnel. Please amend accordingly otherwise it convey different message

Throughout the result: you use code to site quotation but sometimes you also mention participants 7, 8 etc. be consistent. I suggest the cod approach is good since there is not meangfull ordered patient for this study. See lines: 240, 278, 286, 306, 374 etc

Sexual impact: it was overt that you receive feedback only from men. Would be good if can also confirm from women. Most importantly, you did not address it in the discussion section. As we can see here in the result patients were confused. A reader wants to know in the discussion section in the how and why part. Please consider it in the discussion.

Throughout the result and discussion section: you boldly use some psychiatric terms as ‘’anxiety, depression….’’ Without diagnosis it is difficult to label patients with such symptoms so better to use appropriate terms such depressive symptoms. See line 405

Page 16: line 436: …….tests, and investigations…does these differ? Check it

Line 446: lack of continuity of care: see my concern above

Discussion

….It is the first study in Ethiopia but

see https://www.tandfonline.com/doi/full/10.2147/RMHP.S443475

https://www.nature.com/articles/s41598-023-47567-x

Better to address sexuality in line with the disease and or medication

Better to consider existing healthcare financing issue with the patient concern (were all of them not member of CBHI)?? If so there is a problem in participant selection

Health perception is not well addressed in the study but present in the discussion line 468

Continuity of care and interprofessional communication should revisit. I am not sure also always resident doctors are responsible alone. Better to introduce the hospital well in the study setting as who is responsible in treating HF patients

Conclusion

You recommend routine mental health screening in conclusion but lack strong data from the result

6. PLOS authors have the option to publish the peer review history of their article (what does this mean?). If published, this will include your full peer review and any attached files.

Reviewer #1: No

Reviewer #2: **Yes: **Nurilign Abebe Moges

---

## [Author Response · Author response to Decision Letter 0]

2 Aug 2024

Point by point responses to reviewers’ comments. 

Dear editor and reviewers 

Following the review of our manuscript, PONE-D-24-12608, titled “The experience of people living with heart failure in Ethiopia: a qualitative study”, we are pleased to submit the revised version and our responses to comments and suggestions. We would like to thank editors and reviewers for their constructive feedback. 

Reviewer 1

1. Title: I would suggest the title need to be modified ‘Experience of people living with heart failure in Ethiopia: a qualitative study’.

Response: After careful consideration, we have decided to retain the current title as the use of "The" is intentional, as it emphasises our focus on a collective or overall understanding of these experiences. Its presence makes the title more specific in signalling to the reader that the study will cover the overall experiences of people living with heart failure in Ethiopia. 

2. #Abstract: *Background. I would suggest using full wordage for heart failure in line 33, page 2, as abbreviations are not recommended in an abstract, as all abbreviations should be defined in the background section. Therefore, it is best to remove all abbreviations in brackets in the abstract and only use full wordage. Line 37, page 2 you can write as ‘A descriptive qualitative ….’

 Response: These corrections have been made. 

3. #Abstract: * Result: I would suggest that to increase the transparency and attractiveness of the manuscript. In the results report, the number and list of categories, themes, and sub-themes should be indicated in both the abstract and the text. And the authors should also enclose the themes and subthemes in the quotation marks across the manuscript.

Response: We have provided the list of themes in the abstract but due to word limitation, we cannot include the subthemes. 

4. #Abstract: * Line 53, page 2, “People living with HF in Ethiopia experience various symptoms.” What are various symptoms? Based on the authors manuscript findings, the above sentence doesn’t quite inline.

Response: ‘Various symptoms’ refers to a range of clinical manifestations that individuals with heart failure may encounter, including physical, emotional, and other general symptoms. This symptom experience is one of the findings (theme 1) of this research.

5. #Abstract: * Line 53-55, page ‘The impact of HF on various aspects of their lives, combined with poor general health perceptions, significantly affects their HRQoL.’ I could not find any mentions ‘…. poor general health perceptions in your findings. Could you clarify whether it is a health care providers or patients who have these perceptions.

Response: The necessary corrections have been made and the statement revised (Line)

6. #Abstract: * Line 55, page 2, ‘They’ represents ……

Response: ‘They’ refers to the research participants (people with heart failure)

7. #Abstract: *Line 56 and 57, page 2, I could not find any mentions ‘……to understand their experiences and impact on their daily life’ in your findings. * …what are ‘Effective multimodal interventions….’

Response: These details are included in the text. We could not present them in the abstract due to word restrictions. Effective multimodal interventions are combinations of interventions such as psychosocial, pharmacological, educational, lifestyle changes that target various aspects of HF to improve HRQoL.

8. #Abstract: I suggest the authors modifying conclusion based on their pertinent findings. 

Response: We have modified the conclusion.

9. Background: *In the first sentence you explained the prevalence in developed, whereas in the same sentence you discussed quality data in developing countries. It needs to be rewritten.

Response: The sentence has been rephrased for clarity.

10. Background: * Line 66, page 3, ……reason for hospitalisation. It needs to be cited. * Line 67, page 3, …… within family and community. It needs to be cited.

Response: Appropriate references have been cited for both concepts (reference 3 and 4). 

11. Background: * Line 71, page 3, …… unpredictable nature of the disease. It needs to be cited. Line 74, page 3, …… and palpitations. It needs to be cited.

Response: The first two sentences of the second paragraph are taken from references 7 and 8, which are cited appropriately.

12. Background: * Line 75, page 3, …as depression and anxiety. It needs to be cited.

Response: The appropriate reference has been included.

13. Background: *Line 76, page 3, I would suggest using full wordage for HRQoL, the first time it appears in the background section.

Response: This correction has been made.

14. Background: *The fourth paragraph needs to be merged with the second paragraph, as both paragraphs explain HF symptoms and HRQoL.

Response: As the fourth paragraph is about the study’s justifications and significance, it has been left separate. 

15. Background: *Added a little bit HF and HRQoL low resource settings and Ethiopia.

Response: This has been added and incorporated in paragraph 1 for better flow. 

16. Background: *Line 88, page 3…….in Ethiopia (18, 19). Both references do not explain the HF is a significant problem in Ethiopia. what study gaps do authors want to uncover?

Response: More current references have cited. We aim to understand the impact of HF on various aspects of life and how people with HF manage their symptoms and cope with their condition.

17. Methods: *Who distributed invitation letters to all eligible participants?

Response: Invitation letters were distributed by the research assistants during the earlier phase of the project (quantitative survey).

18. Methods: *How to compensate participants for their time when the authors invite them to participate the study?

Response: Our study adhered to ethical guidelines emphasising voluntary participation without compensation. Instead, we expressed our gratitude through thank-you notes.

19. Methods: *Participants were recruited using a purposive sampling technique. according to what criteria? *How did you employ this purposive sampling technique? with who? By whom?

Response: Participants were recruited based on specific inclusion criteria (adults aged 18 years or older with a clinical diagnosis of HF made by a specialist medical doctor, receiving regular follow-up care at a cardiac outpatient clinic for at least three months) and participants characteristics (gender, disease severity, age, and geographical location).

20. Methods: *…. interview guide developed using the TSM. Who was the team that developed it?

Response: The Theory of Symptom Management (TSM) was primarily developed by Dr. Marcia G. Van Eerd and colleagues as part of their research on symptom experiences and management in patients with chronic illnesses. The interview guide used in this study was developed by all authors using the TSM and the results of prior quantitative survey.

21. Methods: *Signs of data saturation were observed after 11 interviews. You used descriptive qualitative approach and used ‘data saturation’ rather than ‘information power’ for sample size determination. ‘Data saturation’ is recommended for grounded theory, but ‘information power’ is recommended for descriptive qualitative approach. Therefore, you should a little bit justify it.

Response: We have accepted your recommendation, and the statement has been revised.

22. Methods: Q#4 is a leading question *The interviews were imported into NVivo version 12 ‘statistical software’ for analysis?

Response: This has been corrected.

23. Methods: *Creating new themes inductively, reviewing themes, defining, and naming themes, and producing the final report (31). Use the updated reference and phase.

Response: This has been undated.

24. # Results: *Were participants given a chance to review their transcripts/ summary of findings?

Response: We attempted to provide transcripts to participants however we are not able to contact all as some had changed their contact details. 

25. # Results: *Line 207, page 8, it is not clear to me ‘Experiences of people living with heart failure’.

Response: This is a subheading in the results section. 

26. # Results: *Authors need to write themes and subthemes as word enclose quotation mark rather than number.

Response: This change has been made throughout the text to make it easier to differentiate between themes and subthemes.

27. # Results: *Themes are broadening idea some are overlapped. It can be merged. *Perception of HRQoL and its influencing factors … needs revision.

Response: Some themes names are revised on the basis of this suggestion. 

28. # Results: *It is important for ‘Initial reaction to diagnosis’ subthemes is resynthesis and see quotes.

Response: While the quotes are concise, they are directly relevant to the theme. However, we appreciate that longer quotes can provide richer context and a deeper understanding of participants' perspectives.

29. # Results: *Line 232, page 10, …. Most of the participants. There is one quotation, how do say most of the participants? *The same issue …. line 248, page 10… Many participants…….

Response: While only one quotation is presented, it encapsulates a common perspective shared by many participants. We only provided quote due to the word limit. 

30. *Line 446, page 17, ‘Lack of continuity of care’. It needs revision.

Response: Continuity of care refers to the consistent and coordinated delivery of medical services to a patient over time. Participants described challenges they faced in follow-up care, including inconsistencies of seeing the same physician and poor communication. We revised the subtheme description and incorporated an additional quote ensure appropriate use of the term.

31. # Results: *Routine mental health evaluations and counselling as part of HF management could be beneficial. How do you come up in this conclusion?

Response: This recommendation could be covered by the first recommendation, so we removed it as the issue was also raised by Reviewer 2.

Reviewer 2

1. The experience of people living with HF…..’’experience’’ is broad term better to indicate patient ‘acceptance of their diagnosis, experiencing symptoms, impact of their quality of life and coping strategy’ since the central theme of the manuscript involves in these concepts. Hence, I strongly recommend the revision of the title by considering such concepts.

Response: While we understand the importance of specifying the key concepts of the study, but we believe that the term " The experience” makes the title more specific and adequately encompasses these themes. Therefore, we respectfully wish to retain the original title.

2. Abstract: Background: (line 35-36) The objective of this study was to explore and describe the experiences of people living with HF and how it affects their health-related quality of life (HRQoL) in Ethiopia. But the title did not indicate the HRQoL, better to consider the title revision above.

Response: The title is intended to serve as an umbrella term covering all themes of the study. We wish to retain the original title to maintain its inclusive nature. The impact of HF on HRQoL is presented in theme 3 and we revised this theme name. The manuscript clearly highlights the specific themes, including HRQoL, to provide detailed insights into the experiences of people living with heart failure.

3. Abstract: Methods: line 37-43: better to indicate the theoretical framework you use in the method section of the abstract.

Response: We have indicated the theoretical framework in the abstract.

4. Results: line 44: You mentioned three themes were identified and three had emerged but what are these them should indicated just immediately to the description. This will give clear idea about the themes to the reader.

Response: The themes have been mentioned in the abstract.

5. Conclusion: …. poor general health perceptions’’ but there is not result indicated poor general health perception the results section of this abstract.

Response: The conclusion has been revised accordingly. 

6. Background: Page 3: line 84: the last paragraph first sentence; ‘’exploring the symptom experiences of people living with heart failure and understanding how they adapt’’ better to your title for this study.

Response: We believe the current title effectively covers all the six themes of the manuscript. Participants spoke about things other than their symptom experience 

7. Background: Line 90-91: ‘’theory of symptom management’’ it is good to indicate why you use this particular theory and give some details just to inform the reader about it. You may use theories either to formulate your study question or do your analysis or both, but it is not clear why you use this particular theory. Please make it clear. I understand you did not fully using its constructs in either case. Any justification please? The question also extends to the method section too,

Response: The TSM was used as it provides a robust framework for understanding the multifaceted experiences of people living with HF, particularly regarding symptom experience, symptom management strategies and impact on HRQoL. This explanation has been incorporated into the methods section. 

8. Methods: Page 5: Table 1: data collection guide: ‘’Demographics’’ but the contents in the right side are not only demographics rather clinical issues,

Response: The question has been revised based on the comment.

9. Methods: Ethical consideration: page 6: line 52-54 …. Permission to conduct the study was also granted from the institutional review board (IRB) of each hospital where the study was conducted. But do these hospitals have IRB? If so better to present the ethical approval number in the similar way done above.

Response: Yes, both hospitals have their own IRBs. The statement has been revised for clarity: “The institutional review board (IRB) of each hospital (SPHHMC IRB Ref No. PM23/9235/13/10/22, SPSH IRB V561/11/10/22) also granted permission to conduct the study.”

10. Methods: Line 161: ‘’Three pretest interviews’’ did you include or exclude in the main analysis? Indicate what do you with these pretest interviews and explain why. Because this is important for the reader and methods section of the qualitative study.

Response: The three pretest interviews were not included in the main analysis. They were conducted to refine our interview guide and ensure clarity and relevance of questions. This explanation as been included in the manuscript. 

11. Results: Page 8: ‘’The age range of the participants was 32 to 73 years, with a mean age of 52.92 years.’’ You use range and mean but not correct statics i.e range with interquartile range while mean is ideal for normal distribution with standard error.

Response: We reported the age range and mean to provide a straightforward summary of our participants' ages. Given the small sample size (n=14) and the studies qualitative design, we used descriptive statistics to contextualise the demographic information.

12. Results: Table 2: the second column in the right end ‘’comorbidity’’ better to designate as ‘’known comorbidity’’ since there may be undiagnosed one.

Response: This has been corrected.

13. Results: Table 3: you mix-up themes which were pre-identified and emerged. Do you have any reason not to use in the ordered fashion?

Response: The order of the themes in Table 3 was presented based on data flow and thematic analysis. We have now indicated in the table the origin of each theme (identified and emergent) providing clarity without altering the natural progression of the thematic analysis. 

14. Results: First theme ‘’ Experience from diagnosis to daily life with HF’’ and its subthemes are not inline. Subthemes state about people’s acceptance and living but experience is very wider term.

Response: The theme has been renamed to “Journey from diagnosis to daily life with HF” to provide more clarity of subthemes.

15. Results: Last theme:’’ Challenges faced in the journey of living with HF’’ and last subtheme ‘’ Lack of continuity of care’’ does not used in the right way. Continuity of care has different meaning in the health system concept but in your study, it was change of healthcare personnel. Please amend accordingly otherwise it convey different message. 

Response: Continuity of care refers to the con

---

## [Decision Letter · Decision Letter 1]

3 Sep 2024

The experience of people living with heart failure in Ethiopia: a qualitative descriptive study

PONE-D-24-12608R1

Dear Dr. Mulugeta,

We’re pleased to inform you that your manuscript has been judged scientifically suitable for publication and will be formally accepted for publication once it meets all outstanding technical requirements.

Kind regards,

Asres Bedaso Tilahune

Academic Editor

PLOS ONE

Additional Editor Comments (optional):

The authors have addressed the issues raised by both authors; therefore, I would like to recommend that the manuscript be accepted for publication. I would like to congratulate the authors for addressing this important area of research and contributing to the scientific literature. Address the final minor revisions suggested by the reviewers.

Reviewers' comments:

Reviewer's Responses to Questions

**Comments to the Author**

1. If the authors have adequately addressed your comments raised in a previous round of review and you feel that this manuscript is now acceptable for publication, you may indicate that here to bypass the “Comments to the Author” section, enter your conflict of interest statement in the “Confidential to Editor” section, and submit your "Accept" recommendation.

Reviewer #1: (No Response)

Reviewer #2: All comments have been addressed

2. Is the manuscript technically sound, and do the data support the conclusions?

Reviewer #1: Yes

Reviewer #2: Yes

3. Has the statistical analysis been performed appropriately and rigorously? 

Reviewer #1: Yes

Reviewer #2: N/A

4. Have the authors made all data underlying the findings in their manuscript fully available?

Reviewer #1: Yes

Reviewer #2: Yes

5. Is the manuscript presented in an intelligible fashion and written in standard English?

Reviewer #1: Yes

Reviewer #2: Yes

6. Review Comments to the Author

Reviewer #1: The authors have done a good job of being responsive to the previous comments. There are, however, a few pending issues that will strengthen the manuscript.

#Abstract

1. The authors can be modified in line 41, page 2 as follows, …. and entered NVivo software for analysis.

2. The following sentence in the conclusion does not clearly present your findings. Please revise it to better reflect your overall findings or consider removing it.

‘People living with heart failure heart failure in Ethiopia experiences various symptoms.’

#Background

The authors cite the following sentence in lines 73 and 74 on page 3, considering it an extension of the idea presented in references 1 and 2. These references describe the epidemiology of HF and CHF and their influence on the social dimension of individuals. Therefore, the authors should include citations that describe HF and HRQoL in low-resource settings and in Ethiopia.

‘It is a significant public health challenge in low-resource settings, including Ethiopia, where it is associated with poor health-related quality of life (HRQoL).’

The same as the underneath sentence in line 74 and 75, page 3, needs to be cited

‘In Ethiopia, HF is a serious issue and the most common reason for hospitalization.’

# Methods

1.In line 136, on page 5, my discretionary comments on your theory of symptom management (TSM) reference paper at https://doi.org/10.1111/j.1365-2648.2009.05179.x that is aligns with your findings and the updated model.

2. In line 131 on page 5, the authors used American English. Please check the entire manuscript to ensure consistency in the use of either British or American English.

#Result

1.In line 253, page 10, the authors should either remove the quote or integrate it into their synthesis paragraph.

2.The following three quotes convey nearly identical concepts, yet the authors have placed them in different subthemes. Please revise it.

In lines 257-259, page 11 “My lifestyle is not as it was before. You should deal with it. I was doing all sorts of activities without difficulty…you know what ...I did everything at home, but now I have limitations, a heavy activity makes me to feel tired” (P11).

In lines 280-282 pages, 11 “I feel down because I can't work as hard as I used to… I become fatigued after working hard…you know ... in that case I feel depressed. I asked God … why is this happening to me?” (P05).

In lines 316-320, page 21 “I feel down when I get tired of doing ordinary activities at home…when I feel 316 depressed, I have nothing to do ...just nothing, I just sit down and take a rest and there is no other solution. I can’t eat what I want. When I see someone else eating something delicious, I become frustrated because I can’t. My life right now is very miserable. I’m angry that I can't go out and come back as I want. This is the worst. I might seem in good physical condition, but my body is weak, which depresses me” (P04).

#Discussion

1…...the current study delves deeper into these issues such as ……describe it ………,

In lines 504-518 on page 19, the themes and subthemes are already described in the results section, creating redundancy. Please remove it.

Reviewer #2: The authors address all comments and questions. I suggest to revisit the concept of ''continuity of care'' otherwise I am satisfied with all response.

7. PLOS authors have the option to publish the peer review history of their article (what does this mean?). If published, this will include your full peer review and any attached files.

Reviewer #1: **Yes: **Zemenu Yohannes Kassa

Reviewer #2: No

---

## [Editor Report · Acceptance letter]

11 Oct 2024

PONE-D-24-12608R1 

PLOS ONE

Dear Dr. Mulugeta, 

I'm pleased to inform you that your manuscript has been deemed suitable for publication in PLOS ONE. Congratulations! Your manuscript is now being handed over to our production team.

Kind regards, 

on behalf of

Dr. Asres Bedaso Tilahune 

Academic Editor

PLOS ONE